# Developing and validating a school-based screening tool of Fundamental Movement Skills (FUNMOVES) using Rasch analysis

Lucy H. Eddy [1,2,3]*, Nick Preston[3,4], Mark Mon-Williams[1,2,3,5], Daniel D. Bingham[2,3], Jo M. C. Atkinson[3,6], Marsha Ellingham-Khan[3], Ava Otteslev[3], Liam J. B. Hill[1,2,3]

1 School of Psychology, University of Leeds, Leeds, United Kingdom, 2 Bradford Institute for Health Research, Bradford Royal Infirmary, Bradford, United Kingdom, 3 Centre for Applied Education Research, Wolfson Centre for Applied Health Research, Bradford Royal Infirmary, Bradford, West Yorkshire, United Kingdom, 4 Academic Department of Rehabilitation Medicine, University of Leeds, Leeds, United Kingdom, 5 National Centre for Optics, Vision and Eye Care, University of South-Eastern Norway, Notodden, Norway, 6 School of Allied Health Professions and Midwifery, University of Bradford, Bradford, United Kingdom

* L.Eddy@leeds.ac.uk

**Data Availability Statement:** Data from all three studies can be viewed on the Open Science Framework (https://osf.io/2uvec/).

## Abstract

### Background

A large proportion of children are not able to perform age-appropriate fundamental movement skills (FMS). Thus, it is important to assess FMS so that children needing additional support can be identified in a timely fashion. There is great potential for universal screening of FMS in schools, but research has established that current assessment tools are not fit for purpose.

### Objective

To develop and validate the psychometric properties of a FMS assessment tool designed specifically to meet the demands of universal screening in schools.

### Methods

A working group consisting of academics from developmental psychology, public health and behavioural epidemiology developed an assessment tool (FUNMOVES) based on theory and prior evidence. Over three studies, 814 children aged 4 to 11 years were assessed in school using FUNMOVES. Rasch analysis was used to evaluate structural validity and modifications were then made to FUNMOVES activities after each study based on Rasch results and implementation fidelity.

### Results

The initial Rasch analysis found numerous psychometric problems including multidimensionality, disordered thresholds, local dependency, and misfitting items. Study 2 showed a unidimensional measure, with acceptable internal consistency and no local dependency, but that did not fit the Rasch model. Performance on a jumping task was misfitting, and

**Funding:** The work of the lead author (L.H. Eddy) was supported by an ESRC White Rose Doctoral Training Partnership Pathway Award (ES/P000745/1). M. Mon-Williams was supported by a Fellowship from the Alan Turing Institute. The work was conducted within infrastructure provided by the Centre for Applied Education Research (funded by the Department for Education through the Bradford Opportunity Area) and ActEarly: a City Collaboratory approach to early promotion of good health and wellbeing funded by the Medical Research Council (grant reference MR/S037527/). L.J.B. Hill, M. Mon-Williams, N. Preston and D. D. Bingham's involvement was supported by the National Institute for Health Research Yorkshire and Humber ARC (reference: NIHR20016). The views expressed in this publication are those of the author(s) and not necessarily those of the National Institute for Health Research or the Departments of Health and Social Care or Education. The funders had no role in study design, data collection and analysis, decision to publish, or preparation of the manuscript.

**Competing interests:** The authors have declared that no competing interests exist.

there were issues with disordered thresholds (for jumping, hopping and balance tasks). Study 3 revealed a unidimensional assessment tool with good fit to the Rasch model, and no further issues, once jumping and hopping scoring were modified.

## Implications

The finalised version of FUNMOVES (after three iterations) meets standards for accurate measurement, is free and able to assess a whole class in under an hour using resources available in schools. Thus FUNMOVES has the potential to allow schools to efficiently screen FMS to ensure that targeted support can be provided and disability barriers removed.

## Introduction

Fundamental Movement Skills (FMS) include locomotor (e.g. running), object manipulation (e.g. throwing) and stability (e.g. static balance) skills [1]. Competency of FMS has been found to be positively associated with children's socioemotional development [2–4], and educational outcomes [5–10]. Additionally, research suggests that FMS play a crucial role in childhood physical activity [11–13], which in turn is associated a wide range of physical, mental, and social well-being outcomes [14–19]. It is therefore alarming that research consistently finds low levels of FMS proficiency [20–23] and physical activity [18] among school-aged children. Moreover, it is likely that the ongoing COVID-19 pandemic will have resulted in fewer opportunities for children to develop FMS [24], as they have been restricted to their home environment for a period of months, possibly with limited access to outdoor space and with no physical education lessons, or scheduled active breaks (e.g. recess). Therefore assessment and monitoring of children's FMS should be recognised as a high societal priority, particularly at the point they begin re-entering formal education.

Currently in the UK, in order to identify children struggling with motor development of any form (i.e. FMS or fine motor skills), parents/carers require a referral by their family doctor / general practitioner (GP) to an occupational therapist or physiotherapist to be assessed for Developmental Coordination Disorder (DCD). Problems at each stage of this referral process result in the needs of these disadvantaged children often being neglected. Mothers from a lower socioeconomic status are less likely to see a GP to discuss their child's development in the first place [25], and even then the process requires parents/guardians to recognise that their child may have less developed FMS than peers of the same age. It is known that parental perceptions of FMS are not always accurate, as they can be influenced by gender [26]. Additionally, waiting times for appointments might influence attendance, as a recent survey of GPs found that the average waiting time for non-urgent appointments was 14.8 days [27]. GP referrals for motor skill assessments also only add to the workload for physiotherapy and OT services, which are already overwhelmed [28].

The issues detailed above are not limited to the UK, with international research showing dissatisfaction with services for children struggling with motor development [29–32]. There is therefore a need for resourceful solutions that enable systematic and efficient assessment of more children's FMS with greater regularity and less referral bias. The Chief Medical Officer in the UK has signposted schools as an ideal place to host such initiatives [28], due to children spending a large proportion of their day in schools, with Physical Education providing a great opportunity to assess and develop motor skills. Having FMS assessments based in schools

would allow for 'universal screening' of childhood FMS, and would therefore enable children an equal opportunity of being identified as needing additional support, reducing health inequalities, and expediting time to support. Previous research has found that collaboration between clinical services, education and families has the potential to identify children struggling with FMS that may not have been identified otherwise [33].

Klingberg et al. recently established criteria that observational assessment tools need to meet in order to be feasible for use in school settings [34]. Two of the guidelines relate to implementation time, in which the authors suggest that (i) assessment tools should last no longer than ten minutes per child, and (ii) that they should measure FMS using less than six items. These guidelines are consistent with teachers reporting that 30–60 minutes is an acceptable amount of time to spend assessing the FMS of a whole class [35]. This is particularly important as prior to the Covid-19 pandemic, teachers were reporting feeling time pressures on teaching 'core' curriculum subjects (e.g. English, Maths and Science) [36], which resulted in a reduction of time allocated to physical education classes [37]. Post-pandemic, it is likely that children will have fallen behind with school work in these 'core' subjects, and thus schools will likely spend a large proportion of time on getting children 'up to speed' with the assessed aspects of the curriculum [38]. Therefore, if schools are going to be asked to universally screen FMS, it is crucial that assessments can be completed quickly and efficiently, so teachers do not feel increased pressure on workload [35]. Additional criteria for feasibility include the resources necessary for schools to be able to implement the assessment tool. One of these criteria is that the assessment tool should utilise equipment that is readily available in schools [34]. A survey of primary school teachers ($n$ = 851) identifying commonly available equipment in schools included: beanbags, chalk, a tape measure or metre ruler, and a stopwatch [35]. It is particularly important to minimize the cost of FMS assessment tools for schools as it is anticipated that, despite the government allocating additional money for schools in the recent budget, the pressure on school budgets will remain [39]. Additionally, proposed resource feasibility criteria suggest that assessments of FMS in schools should be able to be completed in less than six metres of space, or in the corner of a room [34]. A survey of school teachers recently confirmed that over 85% of schools represented had a suitable space this size indoors (87%), and outdoors (98%) [35].

The final feasibility criteria relates to the format of the assessment tool. There are two formats of observational assessments available–product and process-oriented. Product-oriented assessments refer to those which measure the outcome of a movement (e.g. how far a child can run in ten seconds), whereas process-oriented assessments evaluate the way in which a movement is completed (e.g. are a child's knees at ninety degrees to the floor when running). The Klingberg et al. criteria state that school-based assessments should be product-oriented [34], as process-oriented assessments tend to require lengthier training to ensure a comprehensive understanding of the specific phases of each movement, to enable assessors to make a real-time subjective decision as to whether the child is adhering to specified criteria. Such subjectivity has been found to lead to issues with inter-rater reliability [40]. With a lack of specialist P.E. teachers within the UK [41], and FMS training for school staff and thus knowledge [35], it is likely that process-oriented assessments would not fit within feasibility guidelines which states that school-based assessments should be able to be delivered by school staff who have received less than half a day of training [34]. Less intensive training is required for product-oriented assessments, as they have a focus on functional movement, rather than the form of a movement. This type of assessment is therefore ideal for screening purposes, as children that do not have expert movement patterns but are still able to participate in activities will not be flagged as having problems. Despite the relative speed and ease of product-oriented assessments, they do provide teachers with less information about what children are specifically struggling with.

However, for the purposes of a screening programme, they will provide enough information for a referral for a more comprehensive evaluation of their difficulties.

A recent systematic review revealed that a large number of observational assessment tools have been developed to measure the FMS proficiency of school-aged children [42]. Of these, many were deemed to be unsuitable for use in schools when compared with feasibility guidelines [34]. Many well established measures of FMS, such as the Movement Assessment Battery for Children [43], and the Bruininks-Oseretsky Test of Motor Proficiency [44], were both found to cost nearly £1000 to purchase. Additionally, a large proportion of well-established tools take over half an hour to assess each child (e.g. the Test of Gross Motor Development [45, 46])—which does not align with teacher feedback regarding an acceptable time to assess a whole class (30–60 minutes) [35]. Meanwhile, the assessment tools that were more feasible for use in a school setting had little to no peer-reviewed objective evaluation of their psychometric properties. This verification of validity is needed if such tools are to be deployed in school settings. Our research therefore aimed to develop a new, theoretically grounded FMS assessment tool (FUNMOVES) that is both practical for use in schools and has strong psychometric properties, as measured by modern statistical techniques.

## Materials and methods

### Initial development of FUNMOVES assessment tool

An academic working group was established which included experienced academics from the fields of developmental psychology (with expertise in motor development), public health (physiotherapy and occupational therapy) and behavioural epidemiology (physical activity, sedentary behaviour and behaviour change). The group was formed on the basis of the recent feasibility guidelines [34] suggesting there may be a need to either adapt existing assessments, or develop a new tool to be enable universal screening of FMS ability in schools. The working group then (i) conducted a systematic review to assess the validity and reliability of current measures used to assess FMS in school-aged children [42] and (ii) conducted a study assessing the barriers and facilitators to school-based assessments of FMS [35]. The working group reviewed and discussed the findings from the two initial pieces of work and the feasibility guidelines paper [34], along with their own expert opinion, and decided that a new assessment tool should be developed. Five essential criteria for the new assessment tool were agreed based on relevant literature [34, 35, 42] and their own expert opinion. The criteria were that the assessment tool needed to: (i) be a product-oriented assessment which measures all three aspects of FMS (locomotion, object control and balance); (ii) assess a class of 30 children with only two members of staff within the timeframe of a PE lesson; (iii) be teacher-led (after a short training session) and not require a health professional to be present; (iv) use equipment available in schools (beanbags and chalk), or cheap materials (e.g. electrical tape); and (v) be able to be completed in a small ($< 6$ metres squared), protected space if necessary. The number of items in an assessment tool was not carried over from the Klingberg et al. criteria [34], as duration of assessment was deemed more important. A number of assessment formats were trialled by the research team, before a five metre squared grid marked out into 25 x 1 metre squares was found to be the most promising option for conducting the FUNMOVES assessment activities. This grid allows a class to be split into five 'teams' (one per five metre 'lane'). Using the grid, five children (one from each team) can be tested simultaneously on each of the activities in turn. FUNMOVES was developed to require two members of staff for testing for a number of reasons: (i) to ensure speed of assessment (ii) to help mitigate behavioural issues and (iii) due to research suggesting that most teachers (~78%) believed that they would be able to find an additional staff member to assist with school-based FMS assessments [35].

## Study design

FUNMOVES was piloted and then evaluated and modified using an iterative process, whereby the structural validity of assessment tool was measured using Rasch analysis after each study. The results of the analysis, and issues raised by implementation fidelity checklists were used to adapt the activities within FUNMOVES to ensure development was theoretically driven. Once the FUNMOVES met the essential requirements for acceptable structural validity (see 'Analysis' section), and implementation fidelity was consistent amongst teachers, the assessment tool was finalised. This process spanned three studies (the initial pilot and two additional studies post-modifications), two of which were planned post-hoc as further modifications were required in order for FUNMOVES to meet essential criteria for structural validity. Ethical approval for this study was granted by the University of Leeds School of Psychology Ethics Committee (reference: PSC-591). The individuals photographed in this manuscript (S1 File) have given written informed consent (as outlined in PLOS consent form) to publish these photos.

## Participants

Headteachers of Primary schools were invited to participate in this three part study. The approach was made through a flyer both directly to head teachers and through links with the Department for Education via the Bradford Opportunity Area. When schools indicated an interest in the project, meetings were arranged with the headteacher (or member of SLT that responded on their behalf) and information sheets, and opt-out written consent forms were sent to parents. All children in participating schools verbally assented on the day of testing. The first four schools to respond favourably were included in this study. All four schools were based in the Bradford District Area. Three of the four schools were located within the 10% most deprived neighbourhoods in the England (Index of Multiple Deprivation (IMD) Decile 1), and the fourth school was located in IMD Decile 6. Sample size estimates were calculated in alignment with guidelines for Rasch measurement [47]. A minimum of 150 participants was required for each round of this study to provide 99% confidence of item calibration within 0.5 logits (the default Rasch linear scale) and ensure sufficient power to ensure measurement stability across samples.

**Initial pilot.** The sample of 331 children (181 male) was recruited from one primary school in Bradford, in which all pupils (Reception–Year 6; children aged 4–11 years old) participated ($m$ age = 8.33 years, $SD$ = 2 years). Prior to testing, teachers were asked whether they thought each child had difficulties with their motor skills. Teachers were given guidance as to what a child with motor difficulties may look like at the training session. Teachers identified 23 pupils as potentially having motor difficulties in the pilot study.

**Study 2.** Three hundred and fifteen children (165 male) participated in round two (n = 315, $m$ age = 8.37 years, $SD$ = 1.83 years). Class teachers identified 45 pupils that they thought had motor problems prior to testing.

**Study 3.** Two schools in Bradford were recruited for the final round of testing, in which year 1–6 participated (n = 421). However, the data from one of these schools was deemed unreliable, due to a lack of engagement in teacher training, and little time being allocated for testing (which led to researchers having to come back to lead and score some of the activities). As the assessment tool was not delivered by teaching staff (the intended purpose of FUN-MOVES) this school was not included in the final Rasch analysis. The final sample size for the third analysis therefore comprised 168 children (70 male, $m$ age = 8.42 years, $SD$ = 1.92 years). Teachers identified five children as having potential motor skill difficulties.

## Materials

All teaching staff who took part in the study were provided with a manual during training (see S1 File for the manual used in study 3) which included (i) what FMS are and why they are important, (ii) instructions on what materials were needed (25 beanbags, a tape measure or a metre ruler and chalk or electrical tape) (iii) how to run and score each activity, and (iv) score sheets for each activity. Score sheets asked teachers to record additional demographic information including gender, dominant hand and a judgement for teachers to make prior to testing as to whether they believed each child had motor difficulties. Researchers conducting fidelity check used a checklist which was used to evaluate how accurately teachers implemented FUN-MOVES (see S2 File for the fidelity checklist used in study 3).

## Study procedure

FUNMOVES was evaluated iteratively on 1067 children across years 1–6 (aged 5–11 years), across four schools to collect data for psychometric testing. Reception year (4–5 year olds, $n$ = 48) were also tested in the initial pilot, however, due to issues with attention and comprehension (meaning that FUNMOVES could not be implemented at a whole class level) this year group was not tested in the latter two studies. Prior to testing, teaching staff were provided with an hour-long training workshop in which an introduction on the importance of measuring FMS was given and teachers role-played in interactive sessions to practice instructional and scoring activities. Teachers were encouraged to ask questions throughout the session and were given an email address to contact the researcher after the session. At the end of training, each teacher was given score sheets and asked to group their pupils in groups of five by ability, and fill out the demographic information (gender, date of birth, preferred hand and whether the teacher thought each child had motor difficulties).

Researchers attended the school prior to the start of testing to set up the five metres squared grid, in which one metre squares are used for guiding and scoring the children as they perform physical activities. During the assessment of each class, at least two members of teaching staff were present to score the participants. Teaching staff explained and demonstrated each activity to the whole class. Participants were not permitted to practise. All participants completed one activity before the next was explained, demonstrated and tested. Researchers scored implementation fidelity independently. For the initial pilot, researchers noted any issues they noticed. In studies 2 and 3 an implementation fidelity checklist was used which looked at the number of essential criteria met by teachers to ensure that each activity was run correctly. Researchers corrected teachers if they were implementing activities incorrectly, after noting down issues. After testing, each school was debriefed using reports which detailed how each pupil performed relative to the rest of their year group on each activity, calculated using percentile rank. The same study procedure was implemented for each of the three rounds of testing.

## Analysis

Rasch analysis was used to develop the final FUNMOVES assessment tool, with appropriate modifications to FUNMOVES made after each iteration to enhance its structural validity. Rasch is a form of probabilistic mathematical modelling that has several advantages over classical testing of outcome measures (such as exploratory factor analysis). It determines whether an outcome measure's psychometric properties permit the summing of items' raw scores to provide a total outcome score [45]. In the case of FUNMOVES, the activities form the 'items' of the FUNMOVES evaluation. Moreover, the Rasch approach combines evaluation of a number of psychometric issues such that if item responses (the scores) meet the expected Rasch model,

the summed ordinal scores can be transformed to interval level scaling [45]. Additionally, it enables you to evaluate not only whether all items are measuring the same overarching construct, but also (i) whether there are redundant items in the scale (local dependency) and (ii) how changes to activities (e.g. changes to scoring) may impact the validity of the measure. It is therefore useful when a new scale, such as FUNMOVES, is developed from first principles. Rasch analysis works on the premise that the ability to complete an 'item' is dependent on (i) the difficulty of the item and (ii) the ability of the participant [48]. It uses an item-response model to evaluate participant ability and item difficulty on a shared continuum (logit scale) [49]. Items positioned high on the logit scale are more difficult and individuals high on the scale are more capable. Rasch analysis uses the logit scale to assess the psychometric characteristics of assessment tools [50]. The Rasch analyses in these studies were conducted on each school's item responses that were gathered using the procedure outlined above. The analyses used the unrestricted partial credit model in RUMM 2030 software, as responses varied between items [51]. Each Rasch analysis generates summary statistics including mean 'person' and 'item' locations and a chi squared test indicating fit to the Rasch model. A non-significant chi-square value would indicate no difference between scores expected by the model and those observed in testing, and would suggest that items were measuring consistently across different ability levels [52]. Internal consistency values are also calculated using a 'person separation index' (PSI). An assessment tool which has the ability to differentiate between two or more groups of ability should have a PSI value of $\geq$0.7 [53].

Analyses for individual items (i.e. each activity within FUNMOVES) included fit to the Rasch model (measured using chi-squared and fit residuals), response category thresholds, item response bias (Differential Item Functioning- DIF), and response dependency. Unidimensionality was assessed using principle component analysis which identified the two most divergent subsets of items within the first factor [54]. Person estimates for each of the two sets of items were calculated, and differences between these estimates were assessed using t-tests. For a measure to be classified as unidimensional, there should be no more than 5% of significant tests, or the lower bound of the binomial confidence interval should be less than 5% [52]. Rasch analysis is a more accurate and comprehensive measure of structural validity than factor analysis [55] and has been used previously to validate motor skill measures [56–60]. In the case that FUNMOVES was not multidimensional or had response dependency, items were removed. To ameliorate disordered thresholds, two or more adjacent response categories may be combined. To evaluate the external structural validity of FUNMOVES, in each study an ANOVA was conducted using mean logit scores to see whether there were significant differences between school year groups, genders, and whether or not teachers thought each child had motor difficulties prior to testing.

## Results

### Pilot study

**Activities evaluated.** A description of the activities in FUNMOVES and how they were scored for the pilot study can be seen in Table 1.

**Implementation fidelity.** The most problematic activities with regards to implementation fidelity were static balance and walking along the line, for which researchers noted that there were issues with comprehension (both children and teacher) and scoring of the activities. For static balance, the teacher from one class continually demonstrated the activity whilst each group was being tested, which meant that children were getting to practise and had multiple testing opportunities. Additionally, teachers expressed confusion regarding left and right leg balances; they were not clear whether the leg specified was the one that children should be

**Table 1. A description of the activities included in the first version of FUNMOVES, and how they were scored by teachers.**

| Item | Activity | Scoring |
|---|---|---|
| Running[L, B] | Children run from the first line on the grid, to the back line and back as many times as possible in 15 seconds, touching both lines with their foot. When the teacher says 'STOP' children stop and sit down. | Full lengths and box sat in (converted to metres run) |
| Jumping[L,B] | Children do as many jumps as necessary to stop on the first line of the grid and stop still. The teacher counts for 3 seconds out loud and then sets them off to the next line, where the process is repeated until the back of the grid. | 1–6 –the box where they couldn't complete the activity as instructed (6 for completion) |
| Hopping[L,B] | Children do as many hops as necessary to stop on the first line of the grid and stop still. The teacher counts for 3 seconds out loud and then sets them off to the next line, where the process is repeated until the back of the grid. Activity is completed twice (once on each leg) | 1–6 –the box where they couldn't complete the activity as instructed (6 for completion) |
| Throwing[OC] | Children have 5 beanbags and try to throw one in each box in their lane (underarm). The activity is completed twice (left and right hands). | 0–5 (number of boxes filled with a beanbag) |
| Kicking[OC, B] | Children have 5 beanbags and try and kick (along the floor) one in each box in their lane. The activity is completed twice (left and right feet). | 0–5 (number of boxes filled with a beanbag) |
| Static Balance[B, OC] | Children hold five balance positions, whilst passing a beanbag around their body three times<br>1. Feet shoulder width apart<br>2. Feet together<br>3. Right leg<br>4. Left leg<br>5. One leg eyes closed | 0–5 (number of balances successfully completed) |
| Walking along the line[L,B] | Children walk along the line on the left edge of the grid (which has half meter markings) heel-to-toe, placing one foot in front of the other with no gap | 1–11 (zone where the child can't complete the task as instructed– 11 for completion) |

NB

[L] = locomotor skill

[OC] = object control skill

[B] = balance skill.

balancing on or holding up. For walking along the line, teachers set more than one child off at once and losing track of scoring. Children were also not walking heel-to-toe even when prompted, this sparked confusion amongst teachers about how much leeway they should give children when scoring. Additionally, for the jumping and hopping activities it was apparent that the way children were doing the activity was not standardised, and that some children were doing multiple small jumps/hops between the lines and others were doing one big jump/hop from line to line (making the activity more difficult).

**Rasch analysis brief outline.** The initial Rasch analysis revealed internal consistency below the accepted level ($PSI$ = .68), and misfit of FUNMOVES item responses to the Rasch model ($\chi^2(40) = 108.03$, $p < .001$). Items displaying misfit to the Rasch model were: running ($F(4,318) = 6.10$, $p < .001$); non-dominant leg hopping ($F(4,307) = 5.36$, $p < .001$); and static balance ($F(4,320) = 7.73$, $p < .001$). Five items displayed disordered thresholds–jumping, hopping (both dominant and non-dominant leg), non-dominant leg kicking, and walking along the line. There was also evidence of item response bias for running ($F(6) = 5.41$, $p < .001$), jumping ($F(6) = 6.78$, $p < .001$), static balance ($F(6) = 6.63$, $p < .001$) and walking along the

**Table 2. Rasch analysis descriptive statistics.**

| Analysis | Item location | | Person location | | Item fit residual | | Person fit residual | | Chi-square interaction | | | Person Separation Index (PSI) | | Unidimensionality | | | |
|---|---|---|---|---|---|---|---|---|---|---|---|---|---|---|---|---|---|---|
| | *m* | SD | *m* | SD | *m* | SD | *m* | SD | Value | df | p | With Extrms | No Extrms | Number of sig tests | Out of | % | Lower 95% CI |
| Study 1 | 0 | .51 | .16 | .29 | .70 | 1.09 | -.11 | .96 | 108.3 | 40 | < .001 | .68 | .69 | 32 | 323 | 9.64 | .07 |
| Study 2 | 0 | 1.24 | .98 | .73 | .15 | 1.23 | -.26 | .89 | 45.17 | 28 | .02 | .71 | .71 | 14 | 325 | 4.31 | .02 |
| Study 3 (initial) | 0 | .87 | .75 | .64 | .17 | .86 | -.22 | .90 | 19.56 | 14 | .14 | .67 | .67 | 11 | 168 | 6.55 | .03 |
| Study 3 (rescore) | 0 | .95 | .68 | .75 | .13 | .77 | -.24 | 1.02 | 20.42 | 14 | .12 | .64 | .64 | 9 | 168 | 5.36 | .02 |

line for year group ($F(6) = 4.33$, $p < .001$). Additionally, running showed item-response bias by gender ($F(6) = 12.81$, $p < .001$). Correlations between item residuals also identified local dependency for two sets of items: (i) hopping dominant and non-dominant leg ($r = .41$) and (ii) kicking dominant and non-dominant foot ($r = .19$). The assessment tool was also not unidimensional, as 32 of the 323 t-tests (9.64%) were significant. An ANOVA showed that there was a significant difference between the scores obtained by year groups ($F(6,326) = 25.00$, $p < .001$, $d = 1.25$), in which mean logit score increased with each year group, with the exception of year 2 outperforming year 3, and year 5 outperforming year 6. Additionally, there was a difference in mean logit scores between children identified prior to testing as potentially having motor problems, and 'typically developing' children ($F(1,296) = 30.57$, $p < .001$, $d = 1.22$), in which children identified by the teacher as potentially having motor difficulties performed significantly worse on FUNMOVES. Finally, results showed no difference in mean logit scores between genders ($F(1,308) = 1.13$, $p = .29$, $d = .12$). Rasch analysis summary statistics for all three studies are presented in Table 2.

**Modifications.** Unidimensionality improved to 5.68% when hopping non-dominant leg and kicking non-dominant foot were removed (to address local dependency) as well as walking along the line (to ameliorate implementation fidelity). Inputting the running activity as the number of full lengths (5 metres) run, rather than metres run resolved item misfit. These changes were therefore carried forward to the second version of FUNMOVES, and children were subsequently able to choose which leg they would like to hop on, or kick with. Additionally, the rules for the jumping and hopping activities were changed to specify that children must do at least two jumps between each line to standardise the way children complete those activities.

## Study two

**Implementation fidelity.** There was full compliance with essential criteria in nine out of the twelve classes tested. Average compliance across classes was 98.55%. There were issues with instruction-giving and scoring recorded in the remaining three classes (see S1 Table for specific details).

**Rasch analysis brief outline.** Round 2 of Rasch analysis revealed improvements on the version one of FUNMOVES, with the internal consistency increasing to an acceptable level ($PSI = .71$). Additionally, there was no local dependency between items and FUNMOVES was found to be unidimensional, with only 4.31% significant t-tests. However, some psychometric problems remained. Item-trait interaction was significant ($\chi^2(28) = 45.17$, $p = .02$), indicating some misfit to the Rasch model. Additionally, there were three items with disordered thresholds–jumping, hopping and balance (see Fig 1), and jumping also showed some degree of misfit to the Rasch model. There was also evidence of item response bias by year group for both running ($F(5) = 6.07$ $p < .001$) and jumping ($F(5) = 5.82$, $p < .001$), as well as by gender for running ($F(1) = 17.01$, $p < .001$) and hopping ($F(1) = 13.20$, $p < .001$). An ANOVA showed that there was a significant difference between the scores obtained by year groups (F(5,319) =

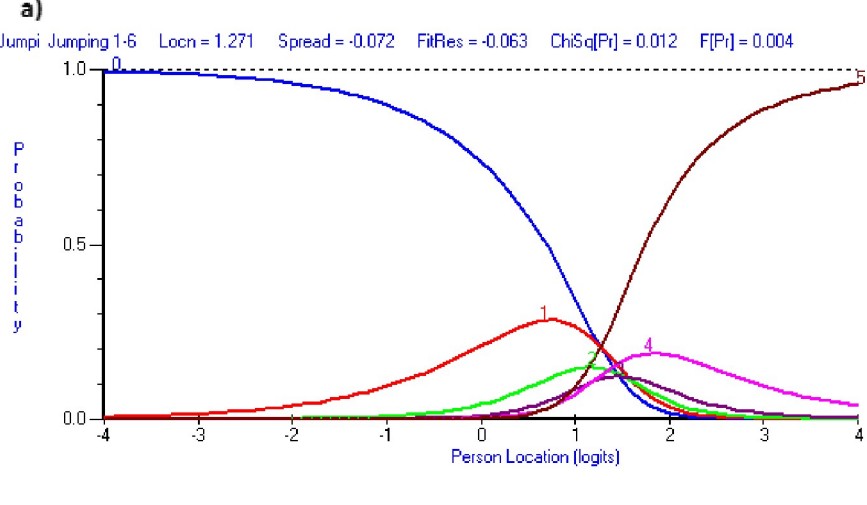

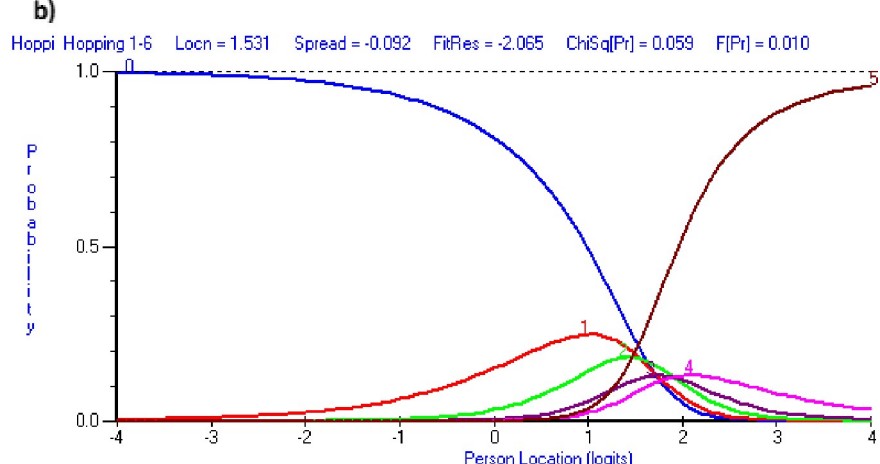

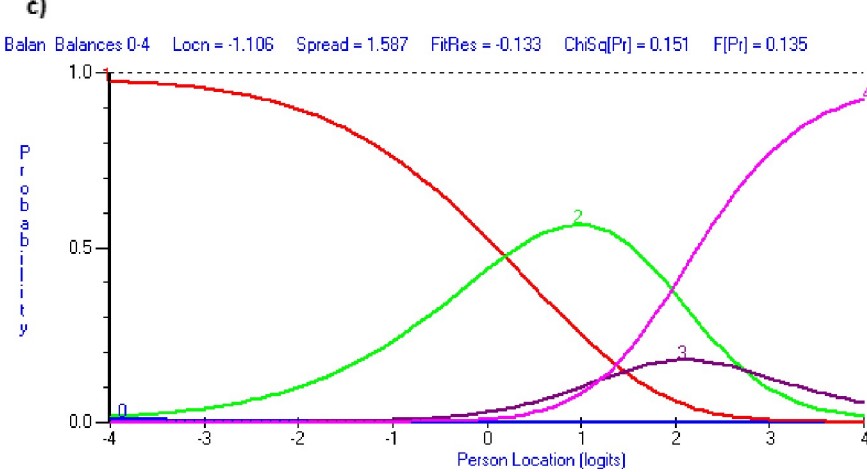

**Fig 1. Category probability curves from round two of testing.** NB: a) shows disordered thresholds for jumping; b) shows disordered thresholds for hopping and c) shows disordered threshold for balance. Graphs were generated by RUMM 2030 software.

53.88, p < .001, d = 1.76), in which mean logit score increased with each year group. Additionally, children identified prior to testing as potentially having motor problems performed significantly worse than 'typically developing' children ($F(1,319) = 9.60$, $p = .002$, $d = .50$). Finally, an ANOVA showed that there was no difference in mean logit scores between genders ($F(1,319) = .06$ $p =. 81$, $d = .03$).

**Modifications.**   As can be seen in Fig 1, the scoring categories for jumping and hopping were not differentiating between abilities. This demonstrates that the 'levels' within these activities did not get progressively more difficult. These activities were modified so that children had to jump or hop to a target zone (marked out in a different colour) on each line. The target zones became progressively smaller, in which the whole of the first line (1 metre wide) was the target zone, and on the final line there was a 10 cm target zone for children to land on. Additionally, Fig 1 demonstrates that children were never more likely to be able to complete balance three than balance four. This showed that the final two balances were in the wrong order for their difficulty level, and were therefore swapped over for study 3.

## Study three

**Implementation fidelity.**   There was full compliance with essential criteria in four out of the six year groups tested, and there were only issues with instruction-giving recorded in the remaining two (see S2 Table for specific details). Average compliance was 96.17% across classes. Researchers deemed the timing and scoring of activities as reliable for all year groups.

**Initial model fit.**   Round 3 of analysis revealed a unidimensional measure (6.55% significant tests; 95% CI = .03, .1) which had a good fit to the Rasch model ($\chi^2(14) = 19.56$, $p = .14$) and just below acceptable internal consistency ($PSI = .67$). Additionally, there were no misfitting items, local dependency or item response bias. Disordered thresholds were found for running, jumping and hopping.

**Modifications and updated model fit.**   The scoring of running, jumping and hopping were modified to ameliorate disordered thresholds (see Fig 2). For running, scores 1–5 were combined as no child was more likely to get 1–5 than 0 or 6. For jumping and hopping scores were changed to: 1—cannot do the activity, 2- can do the activity up to the half way (line 3), 3- can do it past half way but cannot finish it and 4- can complete the activity. These categories were chosen based on the frequency of responses within original scoring categories. Jumping and hopping still presented with disordered thresholds, however, when accounting for 95% confidence intervals, the thresholds were ordered. These modifications improved the unidimensionality of FUNMOVES (5.36% significant tests; 95% CI = .02, .09). Additionally there were no misfitting items, or local dependency. The internal consistency (PSI) was lower at 0.64 than the minimum usually accepted for comparisons between individuals (0.7). Despite this, the person-item map (see Fig 3) enables confidence that this PSI value is acceptable in a screening tool for differentiating between children with age-appropriate motor competence and a group of children with poor motor skills. Item response bias was identified for balance, by gender, however, the differences between boys and girls were minimal and thus the activity was not split. An ANOVA showed that there was a significant difference between the scores obtained by year groups ($F(5,319) = 53.88$, $p < .001$, $d = 2.14$), in which mean logit score increased with each year group. Additionally, there was a difference in mean logit scores between children identified prior to testing as potentially having motor problems, and 'typically developing' children ($F(1,166) = 5.42$, $p = .02$, $d = 1.06$), in which teacher identified children performed significantly worse on FUNMOVES. Analysis also revealed that gender did not impact mean logit scores ($F(1, 419) = .03$, $p = .85$). The final version of FUNMOVES allowed teachers to measure the FMS of a whole class of 30 children in 42 to 58 minutes. An

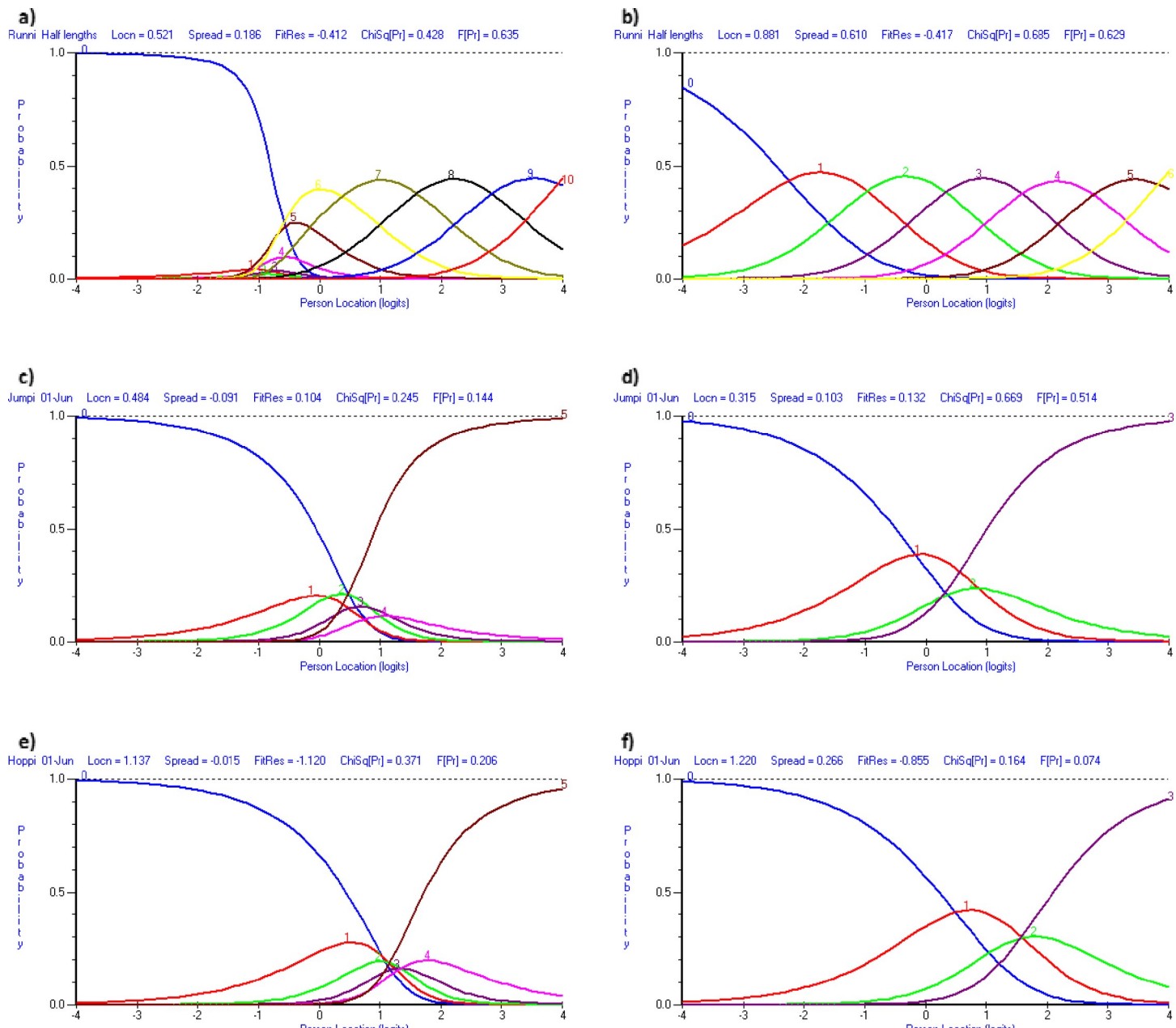

**Fig 2. Category probability curves from round three of testing.** NB: a) shows disordered thresholds for running and b) shows those categories as ordered once scores 1–5 were combined. c) shows disordered thresholds for jumping and d) shows those categories as ordered (within 95% confidence intervals) once categories 1 and 2 were combined and 3 and 4 were combined. e) shows disordered thresholds for hopping and f) shows those categories as ordered (within 95% confidence intervals) once categories 1 and 2 were combined and 3 and 4 were combined. Graphs were generated by RUMM 2030 software.

overview of the changes made to FUNMOVES throughout the development process can be seen in Table 3.

## Discussion

This article describes the development of FUNMOVES, a school-based measure of FMS for primary school children. FUNMOVES is unidimensional, has a level of internal consistency

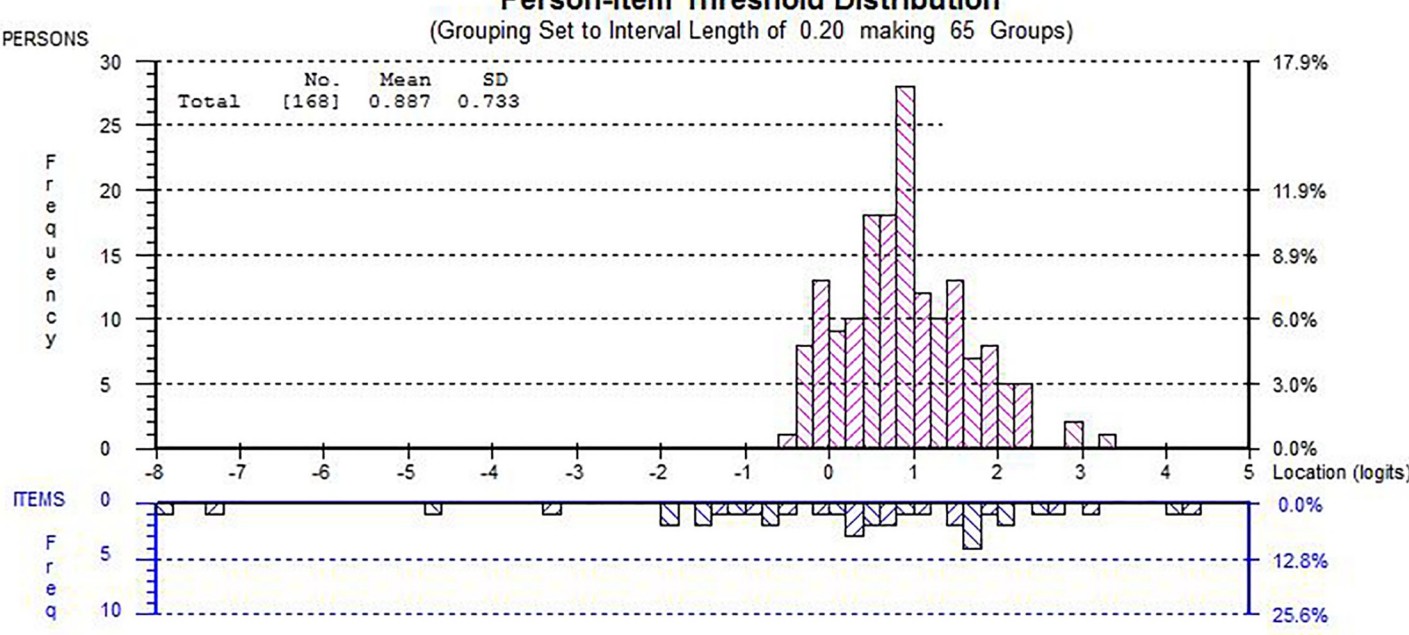

**Fig 3. Person item map for round three of testing.** NB: Graphs were generated by RUMM 2030 software.

which allows for screening of children with poor motor skills, and is able to differentiate between abilities and ages. This is an important development for the assessment of childhood FMS as the current system (only available via health service pathways) is fraught with health-care inequalities [25] and overstretched services [28], and research has demonstrated the merit of school-based initiatives to identify children that the current system fails [33]. As pre-existing more feasible assessments of FMS have been deemed unsuitable for use in schools due to limited or poor evidence for validity and reliability [42], FUNMOVES utilised Rasch analysis throughout the development process to ensure strong structural validity. The sample size in each study enabled a strong calibration of items to the Rasch model [47], allowing confidence in the finalised FUNMOVES battery. Importantly, FUNMOVES was developed to be freely available to schools, unlike many well established measures of FMS, in an attempt to prevent pressure on school budgets influencing a school's likelihood to assess these skills [39].

It is important to consider the feasibility of FUNMOVES for use in schools alongside the results demonstrating structural validity, as it is known from educational research that there needs to be a trade-off between feasibility and validity/reliability in order for school-based initiatives to be implemented consistently and effectively [61]. Klingberg et al. outlined seven criteria that assessments should meet in order to be feasible for use in schools [34]. This research has demonstrated that FUNMOVES meets five of these criteria. As the timing guideline relates to the serial manner of testing that current assessment tools require (less than ten minutes per child), it is not possible to evaluate against this criterion. However, research has established that teachers report class-level assessments that take between 30 and 60 minutes to complete are acceptable [35]. In study three, the final version of FUNMOVES was consistently implemented in under an hour (range from 42–58 minutes), which fits within the guidelines set out for whole class assessment, and would allow for testing to occur within a P.E. lesson [35].

In accordance with feasibility guidelines, FUNMOVES is a product-oriented assessment tool that was conducted within a five metre squared grid, using beanbags, a stop watch, a tape

**Table 3. Overview of the changes made to FUNMOVES throughout the development process.**

| Item | FUNMOVES Version | Activity | Scoring | Implementation Issue(s) | Rasch analysis issue(s) | Modifications |
|---|---|---|---|---|---|---|
| **Running** | 1 | Children run from the first line on the grid, to the back line and back as many times as possible in 15 seconds, touching both lines with their foot. When the teacher says 'STOP' children stop and sit down. | Full lengths and box sat in (converted to metres run) | n/a | Misfitting item ($F(6,310) = 5.11$, $p < .01$) | Score by the number of full lengths completed |
| | | | | | Disordered thresholds—large number of response categories, scores ranged between 10 and 45 metres | |
| | | | | | Item response bias between year groups ($F(6) = 5.09$, $p < .001$) and for teacher identified motor problems ($F(1) = 11.87$, $p < .001$) | |
| | 2 | | Full lengths | | Item response bias between year groups ($F(5) = 6.07$ $p < .001$) and gender ($F(1) = 17.01$, $p < .001$) | n/a |
| | 3 | | | | Disordered thresholds–no child was more likely to score 1–5 than 0 | None–likely sampling issue (high SES) |
| **Jumping** | 1 | Children do as many jumps (feet together) as necessary to stop on the first line of the grid and stop still. The teacher counts for 3 seconds out loud and then sets them off to the next line, where the process is repeated until the back of the grid. | 1–6 –the box where they couldn't complete the activity as instructed (6 for completion) | Lack of standardisation in the way children completed the activity–some were doing one jump line to line, others were doing little jumps. | Disordered thresholds—participants were more likely to score 0 than 1 or 2, and more likely to score 5 than 3 or 4 | Children must do at least two jumps in between each line |
| | | | | | Item response bias between year groups ($F(6) = 7.27$, p < .001) | |
| | 2 | | | Teachers struggled to know how much leeway to give with 'feet together' | Misfitting item ($F(4) = 3.96$, p = .004) | Children jump two footed in a functional way |
| | | | | | Disordered thresholds- participants were more likely to score 0 than 1 or 2, and more likely to score 5 than 3 or 4 | Children jump to a target zone on each line |
| | | | | | Item response bias between year groups ($F(5) = 5.82$, p < .001) | |
| | 3 | + Each line has an increasingly smaller target zone (marked in a different colour) that children have to stop on with both feet | | n/a | Disordered thresholds- participants were never more likely to score 3 than 2, or 4 than 5 | Scoring changed: Can't do the activity Can do the activity up to half way (line 3) Can do it past half way but not finish it Can complete the activity |

*(Continued)*

Table 3. (Continued)

| Item | FUNMOVES Version | Activity | Scoring | Implementation Issue(s) | Rasch analysis issue(s) | Modifications |
|---|---|---|---|---|---|---|
| **Hopping** | 1 | Children do as many hops as necessary to stop on the first line of the grid and stop still. The teacher counts for 3 seconds out loud and then sets them off to the next line, where the process is repeated until the back of the grid. Activity is completed twice (once on each leg) | 1–6 –the box where they couldn't complete the activity as instructed (6 for completion) | Lack of standardisation in the way children completed the activity– some were doing one jump line to line, others were doing little jumps. | Misfitting item (left leg; $F_{(6, 319)} = 4.15$, $p < .01$)<br><br>Dependency between left and right leg ($r = .39$)<br><br>Disordered thresholds– participants were more likely to score 0 than 1 or 2, and more likely to score 5 than 3 or 4<br><br>Item response bias between year groups–left leg ($F(6) = 3.73$, $p < .001$) | Only complete the activity once (leg chosen by child) |
| | 2 | * only completed on one leg | | n/a | Disordered thresholds- participants were more likely to score 0 than 1 or 2, and more likely to score 5 than 3 or 4<br><br>Item response bias between gender ($F(1) = 13.20$, $p < .001$) | Children jump to a target zone on each line |
| | 3 | + Each line has an increasingly smaller target zone (marked in a different colour) that children have to stop on with both feet | | | Disordered thresholds– participants were never more likely to score 3 than 2, or 4 than 5 | Scoring changed: Can't do the activity Can do the activity up to half way (line 3) Can do it past half way but not finish it Can complete the activity |
| **Throwing** | 1 | Children have 5 beanbags and try to throw one in each box in their lane (underarm). The activity is completed twice (left and right hands). | 0–5 (number of boxes filled with a beanbag) | n/a | n/a | n/a |
| | 2 | | | | | |
| | 3 | | | | | |
| **Kicking** | 1 | Children have 5 beanbags and try and kick (along the floor) one in each box in their lane. The activity is completed twice (left and right feet). | 0–5 (number of boxes filled with a beanbag) | n/a | Dependency between left and right foot ($r = .18$) | Only complete the activity once (foot chosen by child) |
| | 2 | * only completed using one foot | | | n/a | n/a |
| | 3 | | | | n/a | n/a |

*(Continued)*

**Table 3.** (Continued)

| Item | FUNMOVES Version | Activity | Scoring | Implementation Issue(s) | Rasch analysis issue(s) | Modifications |
|------|------------------|----------|---------|------------------------|-------------------------|---------------|
| **Balance** | 1 | Children hold five balance positions, whilst passing a beanbag around their body three times<br>6. Feet shoulder width apart<br>7. Feet together<br>8. Right leg<br>9. Left leg<br>10. One leg eyes closed | 0–5 (number of balances successfully completed) | Children were getting the opportunity to practice<br><br>Difficulty with comprehension of left and right leg for younger children | Misfitting item ($F_{(6, 313)}$ = 5.21, $p < .01$)<br><br>All children were capable of completing balances 1 and 2<br><br>Item response bias between year groups ($F_{(6)}$ = 8.54, $p < .001$) | Balance 1 removed from future iterations<br><br>Teachers demonstrate whist all children are sat down<br><br>One leg balance done once (child chooses leg)<br><br>Addition of extra balance |
| | 2 | * 4 balances:<br>1. Feet together<br>2. One leg<br>3. One leg eyes closed<br>4. One leg pick up a beanbag from the floor | 0–4 (number of balances successfully completed) | n/a | Disordered thresholds– participants were never more likely to score 3 than 4 | Switch over the order of balances 3 and 4 |
| | 3 | * 4 balances:<br>1. Feet together<br>2. One leg<br>3. One leg pick up a beanbag from the floor<br>4. One leg eyes closed | | | n/a | n/a |
| **Walking along the line** | 1 | Children walk along the line on the left edge of the grid (which has half meter markings) heel-to-toe, placing one foot in front of the other with no gap | 1–11 (zone where the child can't complete the task as instructed– 11 for completion) | Teachers could not implement or score this activity correctly even with help<br><br>Confusion about scoring (how close together do feet have to be)<br><br>Only allows the testing of one child at once | Disordered thresholds– participants were more likely to score 0 than 1–8, and more likely to score 10 than 4–9<br><br>Item response bias between year groups ($F_{(6)}$ = 4.75, $p < .001$) | Activity removed from future iterations |

measure (all of which can be found within schools [35]) and electrical tape (used instead of chalk as it was longer-lasting), which is cheap to purchase. Additionally, it is a teacher-led assessment, which requires two members of staff to assess the FMS of a class. In order to famil-iarise teachers with FUNMOVES, a one hour training session was provided prior to testing by researchers. The implementation fidelity results from study three revealed that teachers were missing some instructions when explaining the activities, which is likely a reflection of changes that need to be made to the teacher training session and/or the manual to ensure clarity. Despite this, researchers were confident that teachers were explaining and demonstrating the activities well and were capable of accurate scoring, which demonstrates the potential for teacher-led assessments. Where FUNMOVES falls short of the fidelity guidelines is the num-ber of items within the assessment. The final version of FUNMOVES has six items, one more than the guidelines allow to be classified as having 'good' feasibility [34]. However, the results of the analysis demonstrate that there was no redundancy in the assessment, all items fit the Rasch model, and contributed something to the scale. It is also likely that one less item would affect FUNMOVES' capability for differentiating between children of different ages and abili-ties. Due to FUNMOVES being able to assess all six items, within the 'acceptable' timeframe to assess a class, the additional item should not be considered detrimental to the feasibility of the measure for use in schools.

This study has demonstrated that FMS assessment tools that are feasible (in accordance with guidelines) for use in a school setting can also have strong psychometric properties. Results consistently revealed throughout the three rounds that children identified as poten-tially having motor difficulties by teachers prior to testing scored significantly lower than their peers. It is, however, important to note that these results should be interpreted with caution, as the percentage of children being identified in each sample was small (7% in study one, 14% in study two and 3% in study three) which may have inflated the results. Additionally, when pre-paring reports for schools which identified children that were consistently performing below average compared to their peers. It was noticeable that there was a large proportion of children that were missed or misidentified by teachers. A recent review of the literature highlighted that studies assessing the accuracy of teacher questionnaires of motor skills in comparison to physi-cal assessments yielded mixed results [62]. It is likely that children with more 'obvious' motor difficulties will be identified by these methods, however, this study highlights the need for physical assessment in order for all children with difficulties to be identified in a school setting so they can be provided with additional support.

The Rasch analysis for all three studies identified that there was no significant difference between genders on the average performance on FUNMOVES. This is in contrast to a large body of evidence which finds gender differences for FMS [63–68], in which it is often reported that girls perform better on locomotor tasks, and males outperform females on object control tasks. There was no evidence of item-response bias in relation to gender for any of the locomo-tor (running, jumping and hopping) or object control (throwing and kicking) in study three. It is hypothesised that gender differences for object control skills may be explained by socio-cultural factors, for example what children have been exposed to by their family, peers and teachers [69–71], rather than biological factors, such as strength and limb length, as there is minimal difference for these factors between boys and girls until puberty [72]. It is possible that due to the nature of the object control tasks within FUNMOVES, that boys and girls will either have had equal opportunity to practice these skills (e.g. throwing beanbags is common practice within P.E. lessons) or will have been equally likely to find the conditions novel (e.g. kicking practice is normally done with a ball rather than beanbags). Additionally the lack of a significant difference between girls and boys may also be explained by the fact that there was a

lack of younger children in the sample (e.g. 4–5 years old) whereby gender differences are often reported [64, 73].

Despite no difference in object control skills the final version of FUNMOVES did find item-response bias for gender with the balance activity, in which females scored marginally higher in balance than males despite the same overall level of motor competence. Higher competence levels in balance has been seen in the literature previously [74–77]. However, this difference in scores was limited only to children performing the best on the activity (achieving high scores); there was no gender difference in scores for children performing poorly in balance. Thus, as the tool was designed to screen for difficulties, rather than measure children who have sufficient FMS, the activity was not modified. Additionally to gender, SES is known to have an impact on FMS ability, in which children from a low SES are often less proficient [78, 79], with research from low SES areas in the UK showing 18.5% of children had not mastered any of the four FMS measured and 32% had only mastered one [80]. This is important to highlight as the sample in study three was of a higher SES (IMD Decile 6), compared to the other studies (IMD decile 1). It is therefore likely that participants from study three will have had fewer difficulties with FMS, compared to the schools from studies one and two that were situated within the most deprived 10% of neighbourhoods in the UK. This influenced the decision of the working group to not change the scoring of the running activity in the finalised FUNMOVES battery, despite running having disordered thresholds (children not scoring 1–5) in the final study, as it was believed that removing the lower scoring categories would impact upon FUNMOVES' utility for measuring running ability in low SES children.

## Limitations and future directions

One limitation of this three-part study is that the scoring format for jumping and hopping in the finalised version of FUNMOVES have not yet been tested. After two iterations of development using Rasch analysis, the authors did not believe optimal performance in scoring for these two activities was achieved. Changes were made, and after the third Rasch analysis, adaptions were made to improve the response category threshold ordering for jumping and hopping. The fit to the Rasch model after these changes allows confidence that the new scoring categories will be appropriate, however, it will be important for this to be evaluated in a subsequent study. Additionally, in Study 3, the PSI value was lower than .7 (.64), which is widely acknowledged in the literature as acceptable [53]. As can be seen in Fig 3, many of the participants in this sample were above average ability (with average being 0 on the logit scale), and showed little variance in ability. As there are only a few measurement points where the bulk of abilities were located, this explains why the PSI was lower than accepted. The scoring thresholds were, however, spread out sufficiently along the scale range, which suggests that the scale will capture children across the full range of ability and, most importantly, identify the group of children that should be highlighted in screening programmes (i.e. those with poor FMS). It will, however, be important for subsequent research to evaluate responsiveness and on children with a broader range of motor ability, including children known to have poor FMS ability (as measured by well-established measures of FMS ability), particularly as there are a number of issues with asking teachers to identify children with potential problems and then assess children's ability, including that teacher identification of ability is not always accurate [81], and bias due to pre-conceptions of ability.

With regards to measuring ability, it is also important to recognise that FUNMOVES is not validated to identify FMS problems for 4–5 year old children in their first year of formal education (Reception year). For this age group, in study one, testing was completed in groups of five, as suggested by the Reception teachers. This methodology was effective, and allowed children

to complete the activities with minimal confusion. The extra staff necessary to take these children out of their classes and assess them using FUNMOVES was not problematic due to extra support staff being available for Reception in this school. It would, therefore, be interesting for future research to evaluate whether the finalised FUNMOVES battery of activities implemented in this way is valid, reliable, and feasible as it is known that early identification of motor skill problems is beneficial [82]. It is also important to note that this is only a first step in validating FUNMOVES, and only structural validity has been formally assessed to date. More research will be required to evaluate all psychometric properties outlined by COSMIN guidelines [83], as current FMS assessment tools have previously been selective about the psychometric properties measured [42]. Additionally, despite promising signs of feasibility when compared to pre-determined criteria [34, 35], it will be important for future research to establish whether school staff believe that FUNMOVES is practical on a large scale. It will also be important, to evaluate teachers' ability to accurately implement and score the activities without the supervision of a research team, if it is to have utility in a school-based screening programme, particularly as data was deemed unreliable from a school which had researchers present. It is, however, important to note that the unreliable data was not due to issues with teachers implementing or scoring the activities, rather the school not leaving enough time for assessment. Finally, as FUNMOVES is a group assessment, future research will also need to evaluate whether external factors influence performance on activities (e.g. attention) and establish whether the order children are assessed in may play a role in the scores children receive.

## Conclusion

Using FUNMOVES, two members of teaching staff are able to assess the FMS of a whole class in under an hour (following a short introductory training session), in a small space (5x5 metres squared) using items readily available in schools (e.g. beanbags and chalk) or cheap resources (e.g. electrical tape). FUNMOVES therefore has the potential to be used for universal screening of childhood FMS in schools. A more collaborative approach to FMS assessment has the potential to provide further links between healthcare and education, and expedite time to assessment and intervention, which could be vital in response to skill development delays attributed to the Covid-19 pandemic.

## Supporting information

**S1 File. Teacher manual from study 3.**
(DOCX)

**S2 File. Teacher implementation fidelity checklist used in study 3.**
(DOCX)

**S1 Table. Implementation fidelity issues for study 2.**
(DOCX)

**S2 Table. Implementation fidelity issues for study 3.**
(DOCX)

## Author Contributions

**Conceptualization:** Lucy H. Eddy, Nick Preston, Mark Mon-Williams, Daniel D. Bingham, Jo M. C. Atkinson, Liam J. B. Hill.

**Data curation:** Lucy H. Eddy.

**Formal analysis:** Lucy H. Eddy, Nick Preston.

**Investigation:** Lucy H. Eddy, Nick Preston, Daniel D. Bingham, Jo M. C. Atkinson, Marsha Ellingham-Khan, Ava Otteslev.

**Methodology:** Lucy H. Eddy, Nick Preston, Mark Mon-Williams, Daniel D. Bingham, Jo M. C. Atkinson, Liam J. B. Hill.

**Project administration:** Lucy H. Eddy, Nick Preston, Marsha Ellingham-Khan, Ava Otteslev.

**Resources:** Lucy H. Eddy.

**Supervision:** Nick Preston, Mark Mon-Williams, Daniel D. Bingham, Liam J. B. Hill.

**Validation:** Nick Preston, Mark Mon-Williams, Daniel D. Bingham, Jo M. C. Atkinson, Liam J. B. Hill.

**Visualization:** Lucy H. Eddy.

**Writing – original draft:** Lucy H. Eddy.

**Writing – review & editing:** Lucy H. Eddy, Nick Preston, Mark Mon-Williams, Daniel D. Bingham, Jo M. C. Atkinson, Marsha Ellingham-Khan, Ava Otteslev, Liam J. B. Hill.

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
