## [Decision Letter · Decision Letter 0]

13 Jan 2021

PONE-D-20-33626

Developing and validating a School-Based Assessment of Fundamental Movement Skills (FUNMOVES) using Rasch Analysis

PLOS ONE

Dear Dr. Eddy,

Thank you for submitting your manuscript to PLOS ONE. After careful consideration, we feel that it has merit but does not fully meet PLOS ONE’s publication criteria as it currently stands. Therefore, we invite you to submit a revised version of the manuscript that addresses the points raised during the review process.

We managed to secure three reviews for your paper, all of which note the value of this work but at the same time raise important points that need to be addressed. In particular, there may be some issues that may need careful consideration and to be addressed as limitations.

We look forward to receiving your revised manuscript.

Kind regards,

Catherine M. Capio

Academic Editor

PLOS ONE

Journal Requirements:

4. We note that Supporting Information 1 includes an image of a patient.

Reviewers' comments:

Reviewer's Responses to Questions

**Comments to the Author**

1. Is the manuscript technically sound, and do the data support the conclusions?

Reviewer #1: Yes

Reviewer #2: Yes

Reviewer #3: Partly

2. Has the statistical analysis been performed appropriately and rigorously? 

Reviewer #1: Yes

Reviewer #2: Yes

Reviewer #3: Yes

3. Have the authors made all data underlying the findings in their manuscript fully available?

Reviewer #1: Yes

Reviewer #2: No

Reviewer #3: Yes

4. Is the manuscript presented in an intelligible fashion and written in standard English?

Reviewer #1: Yes

Reviewer #2: Yes

Reviewer #3: Yes

5. Review Comments to the Author

Reviewer #1: Thank you for allowing me to review this interesting manuscript. I can see the authors have done lots of work to get to this stage. An easy to use universal screening motor test is important for teachers to find out children who needs help.

My main concerns of the manuscript are as follows:

1. The children who might have motor delay were identified by the teachers, who were also the assessors using the FUNMOVES. This raises a question of bias as the assessors were not blinded. Furthermore, were these children "true-positives" with motor delay or other delay such as in language domain affecting their understanding of the instructions during the test? To test the discriminative ability of the FUNMOVES, it is essential to ensure that the children do have motor delay. I like the authors to comment this and point this out in their Discussion. Another psychometric study on children with motor delay would definitely prove the discriminative power of the FUNMOVES.

2. It is a pity that the FUNMOVES cannot be validated on children aged 4 to 5 years. It is essential to find those children who need help in earlier age than 6 years old due to their neuroplasticity. I understand to test a large of children in a short period of time is important in school settings. I wonder if testing these young children in smaller patches would help. This may be another follow-up psychometric study of the FUNMOVES for the authors.

3. It is uncommon to present the person-item maps in the format of this manuscript. This makes the interpretation quite difficult for the problematic test items. Please consider to use the traditional person-item maps used in Rasch analysis studies.

4. I may have missed this- please state clearly if independent t-tests were performed in each study to test any difference between boys and girls. As mentioned by the authors in the Discussion, it is very unusual that both boys and girls performed equally well in these gross motor skills, especially in young age. Another reason for the non-significant results may be that those aged 4 to 5 years were not included in this study.

5. I understand time is a main factor for a universal screening test. However the difference between dominant and non-dominant legs is very prominent in young children, especially for those with motor delay. However, the FUNMOVES does not explicitly indicate if both legs were tested or the children chose their preferred leg so as to save time. This may not be ideal when understanding child development and children with movement disorders. May the authors comment on this?

Reviewer #2: The manuscript under review describes a study to provide validation information of the FUNMOVES assessment tool. The author employed Rasch analyses in the paper, which seems an appropriate - yet perhaps slightly unusual - method to me. I have offered some comments from reading the paper below. I wouldn't consider myself as an expert in Rasch analyses, I apologize for any comments that I might have raised as a consequence of my lack of knowledge in this area.

1. I'm not knowledgeable in the UK situation, so I'm quite surprised that poor FMS is almost seen as a clinical health issue. I'm a little curious as to (i) whether this is applicable in other parts of the world, and (ii) what kinds of treatments patients would receive when they are referred. That said, these information are probably not extremely relevant to the main contents of this paper, so I'm happy if the authors decide to leave out such information in the paper.

2. There is a large difference between the approach of outcome-based tests (e.g., FUNMOVES) and "progress-based" tests (e.g., TGMD). I think this is a clear difference which some readers may not be aware of. I think the paper would benefit from some discussion and comparison between these methods - effectively, what are the pros and cons of each approach, specifically for the purpose of screening?

3. Can the authors please provide a bit more context in terms of how PE classes are typically structured in UK? By knowing the frequency, duration, class size, number of teachers per class, etc of typically classes would help readers like me understand how feasible the tool can be applied. For example, from where I am from, PE classes are typically led by a single teacher, does that mean this tool may not be applicable? (to be fair, other tools wouldn't work well either)

Also, I am rather curious as to what other students would be doing while the assessment is in progress. Would doing the assessment first/last give them a(n) (dis)advantage (e.g., if they could see what the test are beforehand)? Also, in the longer term, are the test results still valid when students do the test again a year later (assuming the screening is done annually), since they would have time to practice?

4. What are the typical rates of students who meet the criteria for needing treatment in schools (or more specifically low SES schools)? Again, I feel this would provide context as to how "typical" the schools included are.

5. Line 181: could authors be more specific as to what "sufficient power" means?

6. I fully acknowledge the importance of having tools readily available to schools. But I am still a little skeptical as to how similar kicking a bean bag versus a ball would be. The roughness of the floor also greatly impacts how far the bag would travel. Without any practice at all, is this assessment still fair?

7. I believe the final "screening score" is the score by summing all the tests. But since each activity has different scores, is there a need to adjust or scale them? (or is that done by using logit scores?)

8. I personally felt a written description of the modifications done would be helpful - potentially these could be added as additional columns in Table 1? That way it would be easier to see where things started and how they ended. (Figure 3 is not very clear as it doesn't describe the changes made)

9. Given this is a screening tool, can the authors provide any insight as to what threshold should be applied or used for identification of students that need to be referred? For example, can the authors provide the mean scores of students who were reported as having motor problems?

10. I feel the authors could provide some discussion in the merits of using Rasch analyses. For instance, why is it more appropriate than factor analyses in this situation?

Reviewer #3: PONE-D-20-33626 presents the results of three studies that led to the development of FUNMOVES, a school-based assessment of FMS competency. The stated purpose was to develop an FMS instrument that could be feasibly used by teachers in physical education for a full class of children. The paper is comprehensive and is generally well-written. The use of Rasch modeling to examine construct validity is a clear strength. I do, however, have many concerns with both the FUNMOVES instrument and the manuscript.

I have spent a great deal of time contemplating the utility of this instrument. While I agree that feasibility within the school setting is important, I am conflicted whether this instrument provides useful information to the teacher about children with movement difficulties. The insistence upon a process-oriented approach may be more feasible, but does not necessarily provide the teacher with information to plan appropriate instruction – as would a process-oriented assessment. With the current focus on physical literacy in physical education (UK, Canada, USA), an instrument that address motor competence rather than skill seems more appropriate. I fully understand that priorities related to feasibility and process-orientation are likely diametrically opposed.

The study is strong with regards to the use of Rasch modeling to provide evidence of construct validity for the instrument. However, I question the face and content validity of the instrument that proceeds construct validity. That is – does this instrument truly measure FMS competence? The most troubling item is running. With the product-oriented assessment of distance traveled over 15 seconds, the instrument is measuring speed, not motor competence. Speed is highly correlated with a variety of biological factors that do not pertain to FMS competence.

Page 3 – Line 64: Are the authors referring to children with Developmental Coordination Disorder? If so, a clearer statement is needed. More introduction related to DCD and FMS development would also be warranted. If not, it would be useful for the authors to then operationalize “struggling with FMS development”.

Page 6 – Section 2.1: The authors’ self description of the working group does not include anyone with expertise in motor development. Many of my concerns are related to the appropriateness of FUNMOVES to measure FMS competence, which I believe would be shared by others in the field.

Page 8 – Sections 2.3.1, 2.3.2 & 2.3.3 – For each of these sections, the last sentence describes the number of children in each sample identified with motor difficulties/problems. Please include more information on how this was determined.

Page 8 – Section 2.3.3 – How does the unreliable data from the school relate to the overall feasibility of this instrument?

Page 8 – Section 2.4 – How were teachers trained in studies 1 and 2? This section only refers to study 3.

Page 9 – Section 2.5 – What proportion of children in study 1 were in the reception year (4-5yo)? How is this reflected in the information provided in section 2.3.1 and then the results in section 3.1?

Page 10 – Section 2.6 – In addition to the Rasch model and ANOVA, how was feasibility analyzed?

Results

• One overall source of confusion in the results was the modifications section in each study. Does this section reflect changes that were made following the study, that would then be reflected in the instrument used in the subsequent study? For example, does section 3.1.4 describe modifications that were made following study 1 and then used in study 2? If so, I am concerned about the changes made to scoring in section 3.3.3 that led to the final instrument, but were not tested in the field. Even if I am incorrect, more clarity in these sections is needed.

• Table 1 – I was confused by the inclusion of the original FUNMOVES items it Table 1. I believe it would be more useful to present the final version of the instrument within the manuscript. The original version could be included as another supporting information document.

• Fidelity – In each of the three studies, it would be useful to report either a percentage of classes that met full compliance and/or the average compliance (%) across all classes. What threshold did the study team determine was “acceptable” fidelity for a class? For example, was 85% fidelity for class 1 in study 3 sufficient?

• Line 392 – The authors need to make a much stronger for argument for why a PSI below 0.7 is still acceptable for a screening measure. The rest of your construct validity evidence is solid, but this point appears to be arbitrary.

Discussion

• In general, I encourage the authors to soften their discussion of the findings. The consistent point of the discussion is that robust psychometric evidence for FUNMOVES has been provided. However, only evidence for feasibility and construct validity have been provided – and the final PSI less than 0.7 is certainly not evidence of robust or strong validity. There are many other psychometric properties that were not examined that are highly relevant to the utility of this instrument, including test-retest reliability, rater reliability (due to the need to assess up to five students simultaneously), content validity, and criterion validity.

• It is also relevant to discuss the feasibility of FUNMOVES within the context of researcher assistance. In all three studies, the researchers were part of the process that made measurement feasible. An example of this issue is the feasibility data in study 3 where the researcher provided the demonstrations instead of the teacher. True feasibility of teachers using FUNMOVES will still need to be established without outside assistance.

• The analysis shows that FUNMOVES is capable of detecting differences between children that were pre-determined to have movement difficulties by the teacher and those that do not. This is certainly a useful piece of validity evidence. However, no information or evidence was provided for how FUNMOVES can be used to identify children “struggling with FMS development”. Evidence of diagnostic accuracy is critical for the instrument to meet the intended purpose.

• No limitations are provided in the discussion.

Overall

• Please review the full manuscript, including supporting information documents, for editing.

• Please review the reference list for errors and consistency.

6. PLOS authors have the option to publish the peer review history of their article (what does this mean?). If published, this will include your full peer review and any attached files.

Reviewer #1: No

Reviewer #2: No

Reviewer #3: No

---

## [Author Response · Author response to Decision Letter 0]

8 Feb 2021

Response to Reviewers file is attached.

---

## [Decision Letter · Decision Letter 1]

30 Mar 2021

Developing and validating a School-Based Screening Tool of Fundamental Movement Skills (FUNMOVES) using Rasch Analysis

PONE-D-20-33626R1

Dear Dr. Eddy,

We’re pleased to inform you that your manuscript has been judged scientifically suitable for publication and will be formally accepted for publication once it meets all outstanding technical requirements.

Kind regards,

Catherine M. Capio

Academic Editor

PLOS ONE

Additional Editor Comments (optional):

Thank you for your submission. The reviewers recommend publication of your manuscript. One reviewer noted a couple of typo errors; as such, it would be helpful to proofread for any such minor issues.

Reviewers' comments:

Reviewer's Responses to Questions

**Comments to the Author**

1. If the authors have adequately addressed your comments raised in a previous round of review and you feel that this manuscript is now acceptable for publication, you may indicate that here to bypass the “Comments to the Author” section, enter your conflict of interest statement in the “Confidential to Editor” section, and submit your "Accept" recommendation.

Reviewer #1: All comments have been addressed

Reviewer #2: All comments have been addressed

Reviewer #3: All comments have been addressed

2. Is the manuscript technically sound, and do the data support the conclusions?

Reviewer #1: Yes

Reviewer #2: Yes

Reviewer #3: Yes

3. Has the statistical analysis been performed appropriately and rigorously? 

Reviewer #1: Yes

Reviewer #2: Yes

Reviewer #3: Yes

4. Have the authors made all data underlying the findings in their manuscript fully available?

Reviewer #1: Yes

Reviewer #2: Yes

Reviewer #3: Yes

5. Is the manuscript presented in an intelligible fashion and written in standard English?

Reviewer #1: Yes

Reviewer #2: Yes

Reviewer #3: Yes

6. Review Comments to the Author

Reviewer #1: The authors have adequately addressed all my comments. I am glad to see that the authors have significantly toned down the manuscript as a whole.

Reviewer #2: i would like to thank the authors for responding to my comments in the previous round of review. I am happy with the responses made by the authors. I have spotted a couple of points that may require some minor edits. Please see below:

Line 231: There's a typo - "FUMOVES"

Line 340: "in which teacher identified children performed..." should be identified as having motor difficulties?

Reviewer #3: (No Response)

7. PLOS authors have the option to publish the peer review history of their article (what does this mean?). If published, this will include your full peer review and any attached files.

Reviewer #1: No

Reviewer #2: No

Reviewer #3: No

---

## [Editor Report · Acceptance letter]

8 Apr 2021

PONE-D-20-33626R1 

Developing and validating a School-Based Screening Tool of Fundamental Movement Skills (FUNMOVES) using Rasch Analysis 

Dear Dr. Eddy:

I'm pleased to inform you that your manuscript has been deemed suitable for publication in PLOS ONE. Congratulations! Your manuscript is now with our production department. 

Kind regards, 

on behalf of

Dr. Catherine M. Capio 

Academic Editor

PLOS ONE